# Experimental Determination on the Deactivation Kinetics of Residue Hydroprocessing in a Two-Stage Fixed Bed Loaded with HDM and HDS Catalysts

**Kaixuan Chen** [1], **Yankai Cao** [1], **Huihong Zhu** [2], **Hailong Ge** [2,*], **Tao Yang** [2], **Zhiming Zhou** [1] **and Zhenmin Cheng** [1,*]

1   State Key Laboratory of Chemical Engineering, East China University of Science and Technology, Shanghai 200237, China
2   Dalian Research Institute of Petroleum & Petrochemical, SINOPEC, Dalian 116000, China
*   Correspondence: gehailong.fshy@sinopec.com (H.G.); zmcheng@ecust.edu.cn (Z.C.)

**Abstract:** Residue catalytic hydrogenation was carried out in a two-stage downflow fixed-bed reactor system, with a HDM catalyst in the first stage and HDS catalyst in the second one, to mimic the industrial operation. The experiments were run at five sets of temperatures (395 °C (first reactor)/405 °C (second reactor), 400 °C/410 °C, 405 °C/415 °C, 410 °C/420 °C, and 420 °C/430 °C). It was found that the initial stage deactivation was mainly caused by the rapid deposition of coke. Gradually, the deposition of metal sulfides leads to a slow deactivation in the middle stage of operation. A deactivation model that considers the catalyst activity as a function of TOS was proposed and applied to the hydrotreating reaction. The deactivation parameters of HDCCR, HDS, HDNi, and HDV in the two reactors were obtained by fitting the experimental data at the outlet of the two reactors. According to the deactivation curves of catalysts, it is proposed that the deactivation of HDM catalyst is faster than that of HDS catalyst, and multi-bed hydrogenation can effectively increase the catalyst life.

**Keywords:** residue; hydrotreating; deactivation; model; catalyst

## 1. Introduction

Residue hydrogenation reaction includes hydrodesulfurization (HDS), hydrodenitrogenation (HDN), the hydrogenation of conradson carbon residue (HDCCR), hydrodemetallization (HDM), etc. [1–3]. Kinetic modeling on residue hydrogenation during the catalyst activity stable period is important to establish a correlation between reaction rates and process conditions, so as to provide an evaluation of the catalyst activity concerning different residues [4].

However, the steady reaction kinetics do not reflect the plant operation when the catalyst is deactivated. As shown in Figure 1, catalyst activity is decreased during residue HDT processes caused by the deposition of carbon and metal, and undergoes three distinct stages, as follows: (1) initial rapid deactivation at the start of the running (SOR); (2) intermediate slow deactivation at the middle of the running (MOR); (3) final rapid deactivation at the end of the running (EOR) [5–8]. The first stage is mostly due to the coke deposition on the catalyst surface [9–11]. The feed containing highly aromatic and heterocyclic hydrocarbons will show higher coking tendency. In stage two, the gradual decline of activity can be attributed to the accumulation of metal depositions on the catalyst pore surface. Metals, mostly nickel and vanadium, are deposited in the form of sulfides on the outer surface or pores of the catalyst, resulting in the decrease in the specific surface area of the catalyst, the increase in internal diffusion resistance, and the drop of the intrinsic activity of the catalyst [12,13].

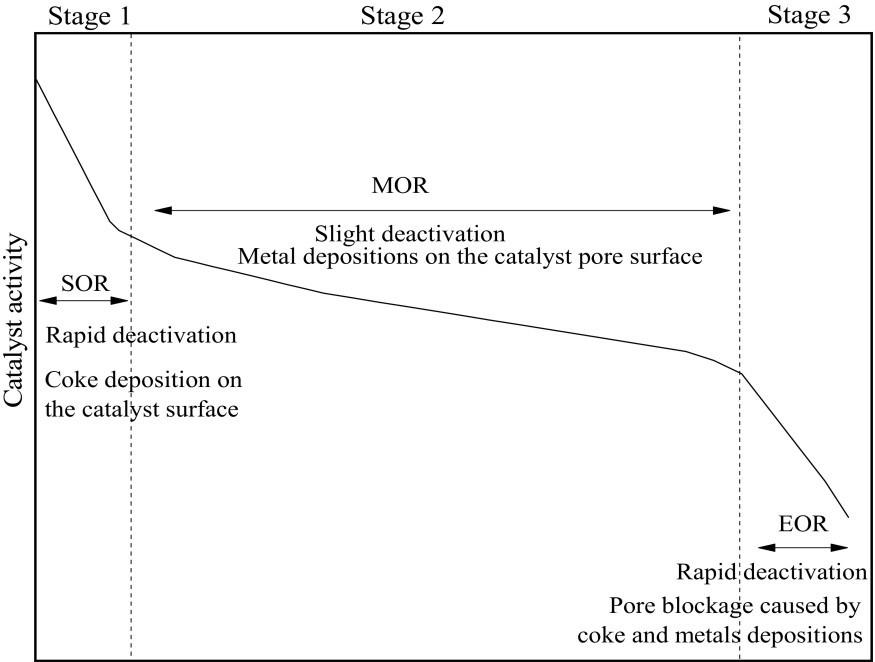

**Figure 1.** Three-stage profile of catalyst deactivation.

Therefore, for residue hydrotreating, a catalyst deactivation model is needed to simulate the entire cycle operation in addition to the steady kinetics model [14]. Additionally, because of the extremely high cost of experiments and the fact that they are time consuming, estimating the performance and life of catalysts under deactivation effects is desirable to simulate the deactivation pattern for HDT reactions for a long time-on-stream (TOS).

There are two main methods for establishing a deactivation model. One takes into account that activity decay is the result of coke deposition, metal deposition, or a combination of both. Another model uses catalyst activity as a function of TOS, because experiments are more accessible and cheaper, and have good predictive performance [15,16]. Various catalyst deactivation models have been reported in the literature [17–21], but there are few studies on the deactivation status and deactivation mechanics for HDM and HDS catalysts packed in series.

The main purpose of this work is to establish residue hydrogenation models of HDM and HDS catalysts, for the optimization of the residue hydrogenation process and prediction of the results. First, two catalysts under different stages of deactivation were analyzed through a deactivation experiment of a long cycle hydrogenation catalyst, and a kinetic model of hydrogenation catalyst is presented. Then, the deactivation model parameters were obtained by fitting the experimental data at the outlet of the two reactors.

## 2. Experimental

### 2.1. Properties of Catalyst and Residue

The HDM catalyst (FEM-10) and HDS catalyst (FES-30) provided commercially were used in this study. Both catalysts are NiMo/Al$_2$O$_3$ type catalysts, and the properties of catalysts are shown in Table 1. The crude oil used in this study is a vacuum residue, and its specific properties are shown in Table 2.

**Table 1.** Properties of catalysts.

| Catalysts | FEM-10 | FES-30 |
| --- | --- | --- |
| Type | Demetallization | Desulfurization |
| Shape | Spherical | Spherical |
| Diameter of particles (mm) | 0.4–0.5 | 0.4–0.5 |
| Average pore size (nm) | 22 | 12 |
| $V_{Pore}$ (mL/g) | 0.7 | 0.613 |
| $S_{BET}$ (m$^2$/g) | 120 | 182 |
| Mo (wt.%) | 6.54 | 1.97 |
| Ni (wt.%) | 13.46 | 2.81 |

**Table 2.** Properties of feedstock.

| | |
| --- | --- |
| API gravity | 9.56 |
| Total sulfur (wt.%) | 5.72 |
| CCR (wt.%) | 22.85 |
| Ni (µg/g) | 43.35 |
| V (µg/g) | 141.5 |
| Distillation (°C) | |
| IBP/10 w% | 349.6/527.0 |
| 30/50 w% | 588.6/641.2 |
| 70/80.8 w% | 706.2/749.8 |

### 2.2. Apparatus

A series of experiments were conducted using several catalysts in a pilot down-flow fixed-bed reactor. (Figure 2) The unit consists of the following three sections: the feed section, reactor section, and separation section. The vacuum residue is preheated and then mixed with hydrogen. The mixture successively passes over the first and second fixed-bed reactors. The first reactor contains 200 mL HDM catalyst, and the second reactor contains 200 mL HDS catalyst. The reactors are heated with electrical resistance wire to ensure isothermal operation. The effective capacity of the isothermal operation section is 1000 mL. In the separation section, the product containing liquid and gas is separated at high pressure and high temperature. The liquid product together with the feedstock will be analyzed.

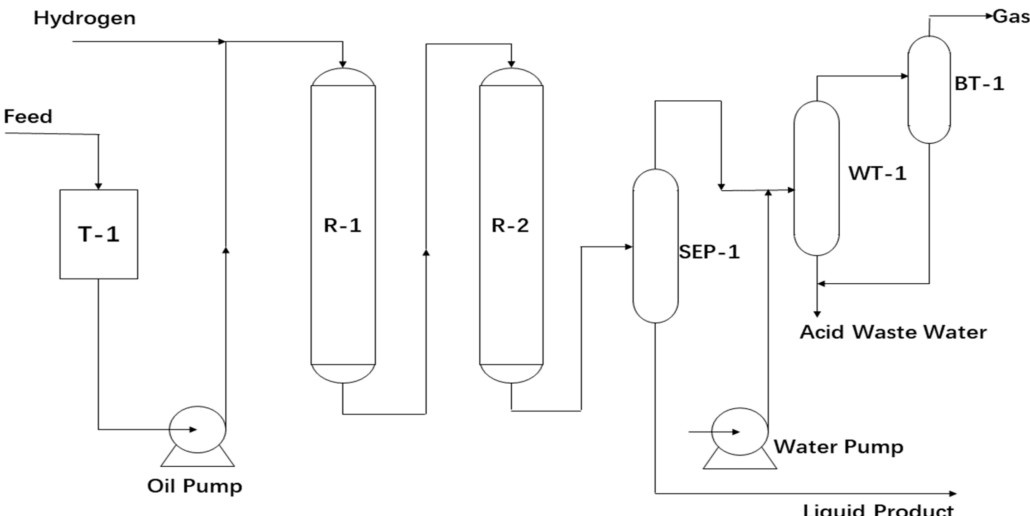

**Figure 2.** Two-stage fixed bed reactor system for residue hydrogenation loaded with HDM and HDS catalysts; T-1, feedstock storage tank; R-1, the first fixed-bed reactor; R-2, the second fixed-bed reactor; SEP-1, high-pressure and high-temperature separator; WT-1, washing tower; BT-1, buffer tank.

### 2.3. Experimental Study on Catalyst Deactivation

The catalysts were sulfided in situ at 230 °C with diesel oil containing 3 wt.% of sulfur. After the catalyst finished sulfidation and the feed oil of the device was shifted to raw residual oil, the temperature was gradually increased to 415 °C for the first bed and 425 °C for the second one, and the pressure was set to 15.0 MPa. The inlet flow rate of raw residual oil and the amount of hydrogen were adjusted to result in a hydrogen–oil volume ratio of 500 *v/v* and a liquid hourly space velocity (LHSV) of 0.8 h$^{-1}$. The device was stopped at 20, 200, 500, and 800 h, and 3 to 4 mL of the catalyst from the top of the bed was discharged for characterization. After that, the remaining catalyst was loaded into the reactor as it is and restarted (no sulfidation is needed at this time). After start-up, the feedstock residue and hydrogen inlet flow rates were adjusted to keep the temperature, pressure, LHSV, and hydrogen–oil ratio constant. After 1500 h of running, the experimental setup was shut down.

In order to understand the deactivation pattern of catalysts during operation, the HDM catalyst and HDS catalyst were characterized using analytical devices, such as a BET device, X-ray diffraction (XRD), inductively coupled plasma–atomic emission spectroscopy (ICP–AES), and an elemental analyzer.

### 2.4. Experiments on Hydrogenation Kinetics

In order to validate the kinetic model of hydrogenation of residue, five sets of experiments were carried out, the specific conditions of which are shown in Table 3, and the composition of the substances exported from the two reactors were obtained. The element analysis of S content was measured by X-ray fluorescence analysis according to the method of ASTM D2622. The V and Ni contents were measured using the inductively coupled plasma (ICP) approach. The amount of Conradson carbon residue (CCR) was measured using the method of ASTM D4530.

**Table 3.** The experimental condition.

| Item | Temperature (°C) | | Pressure (MPa) | LHSV (h$^{-1}$) | H$_2$/Oil Ratio (std m$^3$/m$^3$) |
|------|------------------|------------------|----------------|-----------------|----------------------------------|
|      | 1st Reactor      | 2nd Reactor      |                |                 |                                  |
| 1 | 395 | 405 | 15.0 | 0.8 | 500 |
| 2 | 400 | 410 | 15.0 | 0.8 | 500 |
| 3 | 405 | 415 | 15.0 | 0.8 | 500 |
| 4 | 410 | 420 | 15.0 | 0.8 | 500 |
| 5 | 420 | 430 | 15.0 | 0.8 | 500 |

## 3. Results and Discussion

### 3.1. Characterization of HDM and HDS Catalysts

The XRD patterns of the HDM and HDS catalysts are shown in Figure 3. Both catalysts have strong characteristic peaks at 2θ of 46° and 66.8°, which correspond to the (100) and (110) crystal planes of γ-Al2O3, respectively. In addition, other characteristic diffraction peaks corresponding to γ-Al2O3 were observed at 2θ of 31.9°, 37.5°, 39.4°, and 60.4°, but with the increase in operation time, the peak shape gradually widens, indicating that the crystallinity of the catalyst keeps getting worse during the deactivation process [19,22]. Compared with ICDD/ICPDS standard spectra, the drum at 2θ = 25–27° is mainly amorphous coke peak, which indicates that a large amount of coke has been deposited in the first 20 h. When the running time is about 200 h, the coke deposition is further increased. At the same time, there are sharp peaks at 2θ = 31° and 35° due to the existence of Ni$_3$S$_2$ and V$_3$S$_5$, respectively, which can show that in SOR, the rapid deposition of coke causes the deactivation of the catalyst, and the deposition of metal sulfide also has a certain impact on the initial deactivation. In the spectrum of catalysts with a TOS of 200–800 h, the drum formed by coke deposition did not increase significantly, while the

peak area caused by the metal sulfide increased continuously, which shows that the activity loss of catalyst in MOR is mainly caused by the deposition of the metal sulfide.

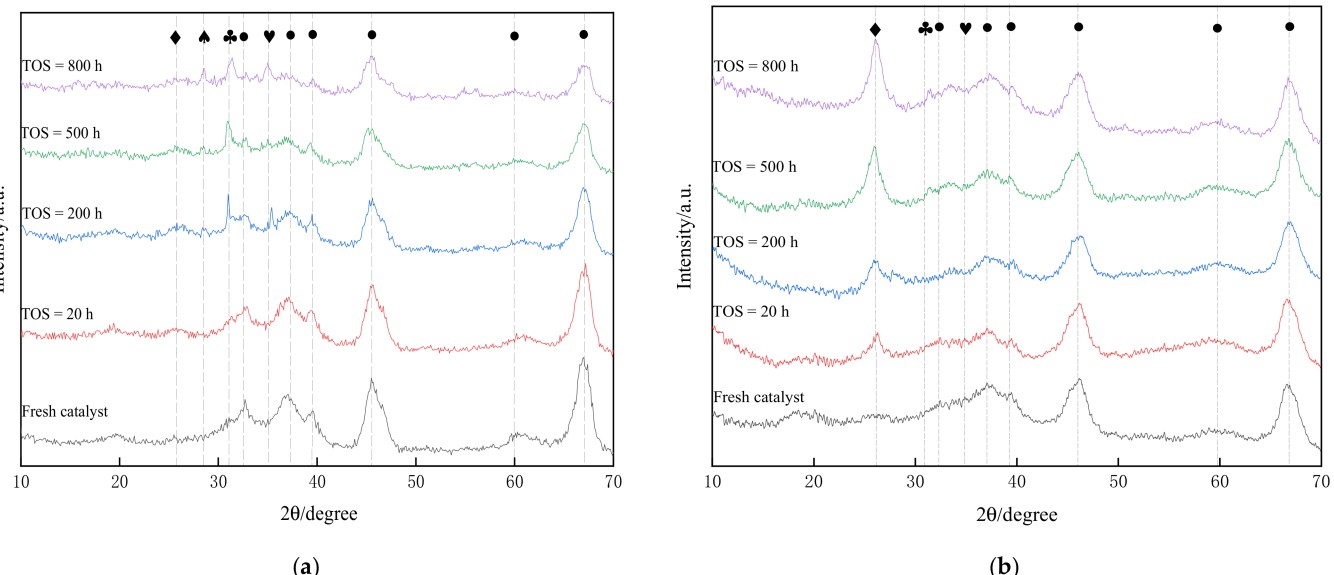

**Figure 3.** XRD spectra of catalysts with different TOS. (**a**) The HDM catalyst; (**b**) the HDS catalyst (• $Al_2O_3$, ◆ coke, ♠ $V_2O_5$, ♣ $Ni_3S_2$, ♥ $V_3S_5$).

It should be noted that the XRD pattern of the HDS catalyst in Figure 3b is significantly different from that of the HDM catalyst. Firstly, the characteristic peak shape of the $Al_2O_3$ of the HDS catalyst under different operating time changes very little and, secondly, the impurity peak caused by metal deposits is not obvious, but the peak shape formed by coke deposition is very obvious. This is due to the fact that after hydrotreating the demetallized catalyst in the first reactor, the feed to the second reactor contains less impurities, especially metallic impurities, which may lead to an insignificant metallic sulfide impurity peak in the XRD spectrum of the desulfurized catalyst and, on the contrary, a more prominent mottled peak was observed by coke deposition. In addition, a small peak of $V_2O_5$ was observed at $2\theta = 28°$ on the spectrum of the HDM catalyst at 200–800 h, which may be due to the oxidation of vanadium during the analysis.

As shown in Figure 4a, the amount of coke deposited on the catalyst increases with increasing operation time. More coke was deposited on the HDS catalyst because of its higher acidity and smaller pore size [6,23]. In addition, a large number of unstable and highly coking intermediates are produced in the first reactor, and coke is easily produced after entering the second reactor with a higher reaction temperature [7,24]. In Figure 4b, a linear relationship can be observed between metal content and TOS, indicating that metal deposits lead to a homogeneous deactivation of the catalyst, unlike coke. As the plant runs and the catalyst deposits increase, the catalyst mass increases, and the metal deposition amount based on the spent catalyst is small. Since the amount of substance of Al in the catalyst remains constant, the amount of metal deposition can be corrected by the amount of Al to obtain the amount of metal deposition based on a fresh catalyst [25]. Additionally, the amount of metal deposition on the HDS catalyst is significantly smaller, further explaining why the heteropeak of metal sulfide in the HDS catalyst XRD spectrum is not evident [26,27]. If the two reactors are considered as a whole, it is observed that less coke and more metal are deposited on the HDM catalyst in the first reactor and more coke and less metal are deposited on the HDS catalyst in the second reactor, which is also consistent with many of the literature reports [8,12,20,28].

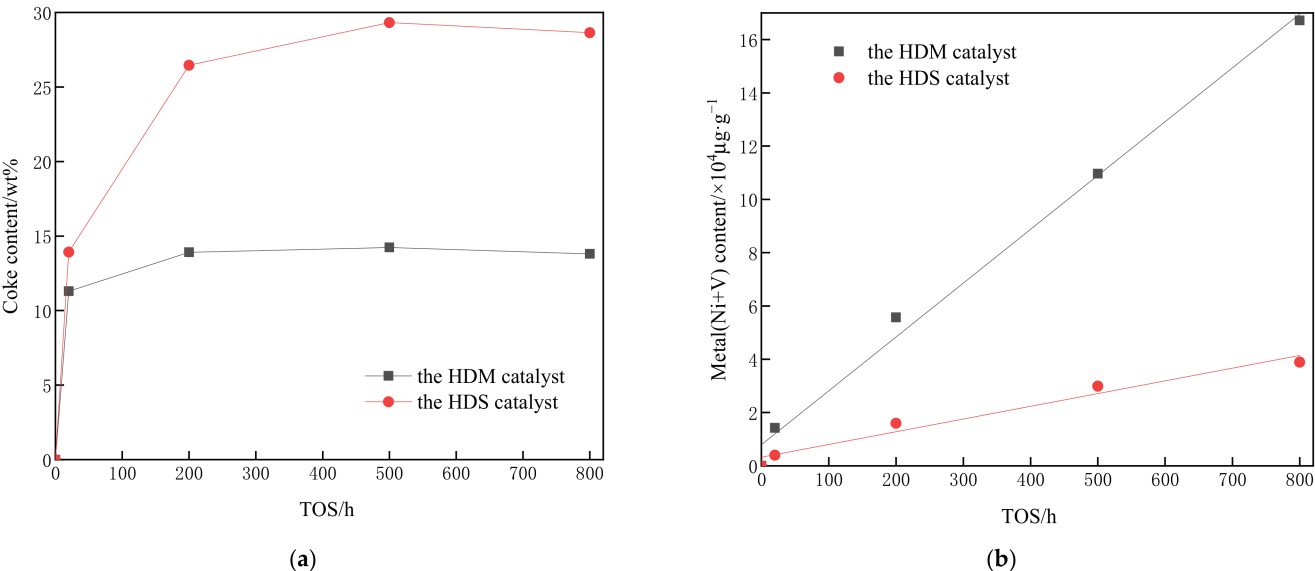

(**a**)

(**b**)

**Figure 4.** Variation of coke and metal sulfide deposition on catalysts with TOS. (**a**) Coke content; (**b**) metal content.

As shown in Figure 5a, the specific surface area loss rate of the HDM catalyst was 16.08% at 20 h and reached 40.56% at 800 h, and the specific surface area loss rate of the HDS catalyst was 38.3% at 20 h and reached 87.1% at 800 h. In Figure 5b, the pore volume loss rate of the HDM catalyst was 30.07% at 20 h and reached 73.32% at 800 h, and the specific surface area loss rate of the HDS catalyst was 42.7% at 20 h and reached 86.5% at 800 h.

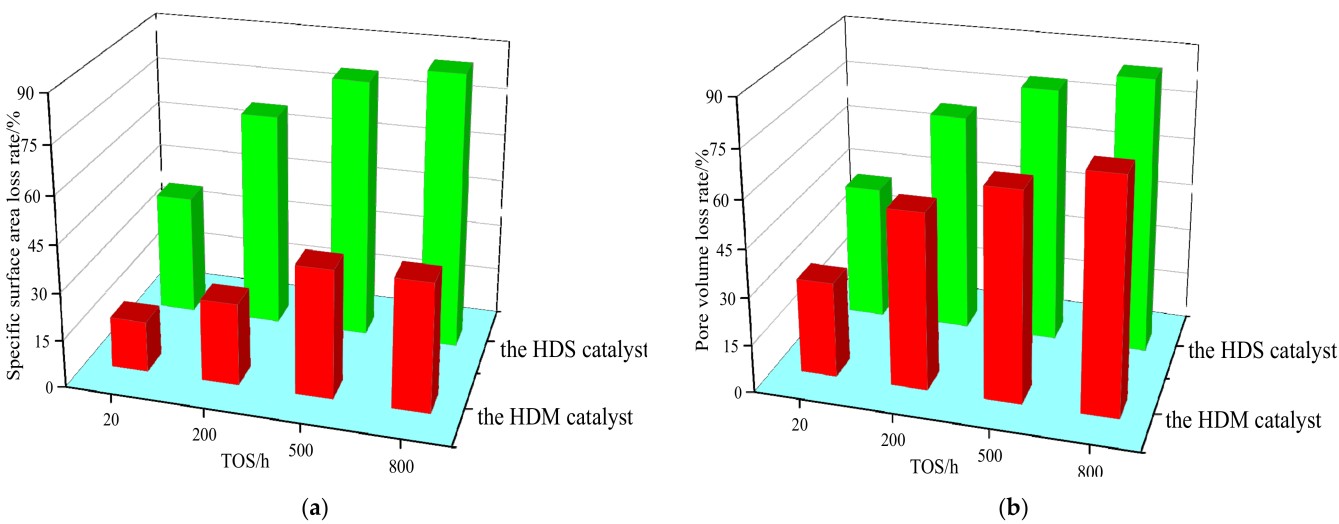

(**a**)

(**b**)

**Figure 5.** Changes in specific surface area and pore volume losses during deactivation of catalysts. (**a**) Specific surface area; (**b**) pore volume.

The rate of change in specific surface area and pore volume loss was significantly higher for both catalysts at 200 h, which indicates that the catalysts were deactivated faster in SOR and slower in MOR. In addition, the specific surface area and pore volume loss rates were significantly higher for the HDS catalyst because more coke was deposited on the HDS catalyst, which had smaller pores, and the deposits were more likely to plug the pores.

Through the analysis of the above characterization results, it is found that the deactivation reasons and rules of the two catalysts are similar. The rapid deactivation at SOR is mainly caused by the rapid deposition of coke, but the effect of metal sulfide deposition at

this stage cannot be ignored; with the passage of operation time, the effect of metal sulfide deposition on catalyst deactivation is more than that of coke deposition, resulting in its slow deactivation at MOR [29].

### 3.2. Deactivation Model of Residue Hydrotreating Catalysts

The catalyst activity $\varphi_i$ of reactions is defined as follows:

$$\varphi_i = \left[\frac{(r_i)_t}{(r_i)_0}\right]_{P,T} \tag{1}$$

where $(r_i)_t$ and $(r_i)_0$ are the reaction rate of reaction i at time t and zero, respectively, as reported in the literature [12,15,26]. In order to calculate $\varphi_i$, the values of $r_i$ at zero time should be measured; however, this is extremely difficult because the experiment cannot obtain the liquid product at $t = 0$. Usually, the initial activities are obtained from the first liquid sample. In this study, however, initial activities are reasonably obtained from the catalyst aging curve by extrapolation, which seems to be more accurate. Accordingly, the catalyst activity at any time can be calculated through the conversion rate obtained by the experiment.

As previously described, catalyst deactivation is due to coking and metal depositions [12,26]. Therefore, the deactivation model considering both coking and metal deposition can be represented by the following equation:

$$\varphi_i = \varphi_i^{\text{coke}} \cdot \varphi_i^{\text{metals}} \tag{2}$$

where $\varphi_i^{\text{coke}}$ is the deactivation function for coking at time $t$, and $\varphi_i^{\text{metals}}$ is the deactivation function for metal deposition at time $t$.

Thus, a new deactivation model incorporating the influence of coke and metal depositions on the whole deactivation process is proposed for describing the catalyst aging curve, as follows:

$$\varphi_i = \frac{1}{(1 + \alpha_i \cdot t)^{\beta_i}} \cdot (1 - \gamma_i \cdot W) \tag{3}$$

The first term of Equation (3) represents the catalyst activity loss due to coke deposition ($\varphi_i^{\text{coke}}$). This activity decline is described as a function of TOS with fitting parameters $\alpha_i$ and $\beta_i$ for reaction i [27]. The deactivation of the catalyst is caused by the deposition of coke and metal, and TOS is not a parameter that affects the deactivation. Correlation with the coke content on the catalyst is the most suitable modeling approach, but multiple sets of experiments are required in continuous operation to obtain spent catalysts at different TOSs, which would make the experiments themselves very expensive and time-consuming. Thus, a model that correlates catalyst activity with TOS is a good choice.

The second term of Equation (3) stands for the catalyst activity drop caused by metal build-up. This activity is related to $W$, which is defined as the ratio between the quantity of metal-on-catalyst (MOC) and the total mass of the spent catalyst (the sum of fresh catalyst and MOC) at time $t$, with a fitting parameter $\gamma_i$. The value of $W$ can be calculated from the metals mass balance as follows:

$$W = \frac{MOC}{MOC + m_{\text{fresh}}} = \frac{\int_0^t [c_{\text{metal, 0}} L_{\text{in}} - c_{\text{metal, t}} L_{\text{out}}] dt}{\int_0^t [c_{\text{metal, 0}} L_{\text{in}} - c_{\text{metal, t}} L_{\text{out}}] dt + m_{\text{fresh}}} \tag{4}$$

where, $m_{\text{fresh}}$ is the mass of the fresh catalyst, $c_{\text{metal}}$ is the metal concentration of the liquid samples, and $L_{\text{in}}$ and $L_{\text{out}}$ are mass flow rates at the inlet and outlet of the reactor, respectively.

By applying the deactivation model to the hydrotreating reaction kinetics, the impurity concentration of the product oil can be predicted more accurately. The influence of external and internal diffusion on the reaction kinetics can be neglected by theoretical calculations of experimental data with small observable parameters for external and internal diffusion. It is assumed that the impurities, such as sulfur, carbon residue, nickel, and vanadium in

residuum are all regarded as a lump and described by the nth order power-law model, and the following equation can be obtained:

$$-\frac{dC_i}{dt} = k_i \cdot \varphi_i \cdot C_i^{n_i} \tag{5}$$

where, $C_i$ is the concentration of impurity lump i, and $n_i$ is reaction order. Additionally, $k_i$ is the apparent rate coefficient of reaction i, following the Arrhenius equation, as follows:

$$k_i = k_{0,i} \cdot \exp\left(-\frac{E_{a,i}}{R \cdot T}\right) \tag{6}$$

Equation (5) was solved by using a fourth-order Runge–Kutta method to obtain the calculated impurity concentration ($C_i^{cal}$) value. Substitute $C_i^{cal}$ and $C_i^{exp}$ into the following objective function, when this function is minimum, and the optimal model parameters can be obtained, as follows:

$$f = \min\sum_{i=1}^{n}\left(C_i^{cal} - C_i^{exp}\right)^2 \tag{7}$$

This objective function was solved using the least-squares criterion with a nonlinear regression procedure based on the trust region algorithm.

As shown in Figure 6, there is a clear linear relationship between $W$ and TOS. Among them, the $W$ values of the HDM catalysts in the first reactor ranged from 7.89 to 10.31 wt.%, and the $W$ values of the HDS catalysts in the second reactor ranged from 1.88 to 2.43 wt.%. The $W$ values varied slightly at different temperatures, but the range of variation was narrow, and it can be assumed that the temperature did not have a substantial effect on the amount of metal deposition for both catalysts. Moreover, the differences in catalyst deactivation due to metal deposition are much smaller at different temperatures, so the effect of temperature on $W$ values is neglected in the deactivation mechanics model, which has been verified in many references [13,28].

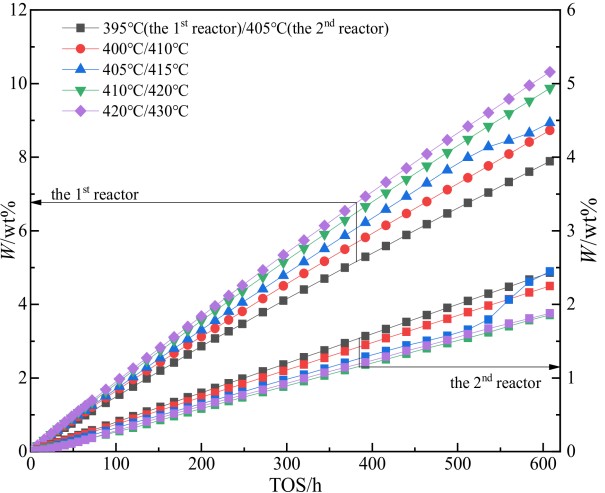

**Figure 6.** The change in $W$ value with TOS at different temperatures (■ 395 °C, ● 400 °C, ▲ 405 °C, ▼ 410 °C, and ◆ 420 °C, line: fitted).

The new model was applied to the hydrotreating reaction kinetics to obtain a set of ordinary differential equations, as in Equations (5) and (6). The parameters of the mathematical model were estimated by regression to match the predicted results with the experimental data, and to optimize the parameter fitting results using sensitivity analysis [30], giving the results shown in Figures 7 and 8. The calculated value of the model is compared with the experimental value, the relative error distribution is shown in Figure 9, and the relative average error of the model does not exceed 5%.

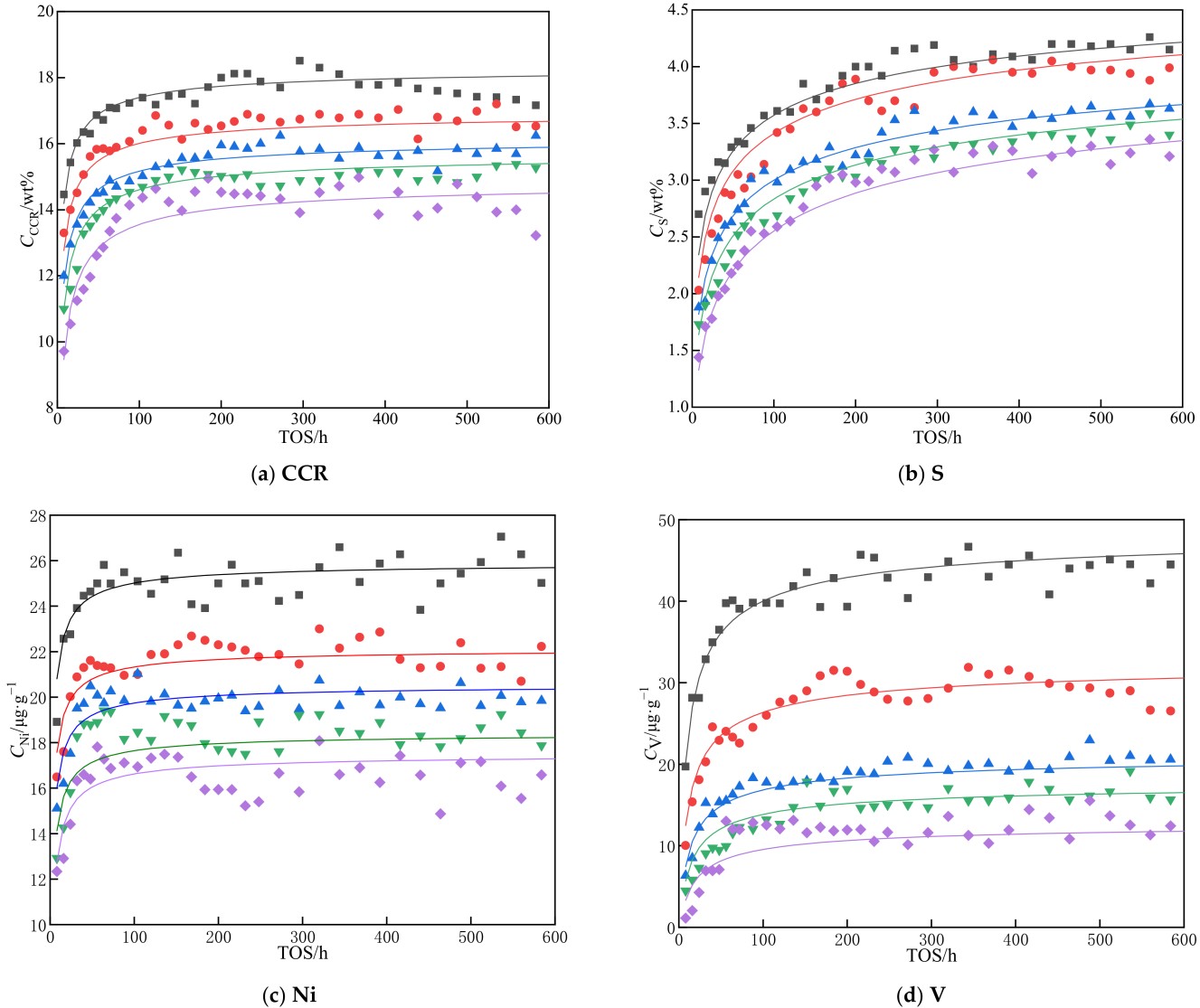

**Figure 7.** Fitting results of the kinetic model for the hydrotreating reaction in the 1st reactor (■ 395 °C, ● 400 °C, ▲ 405 °C, ▼ 410 °C, and ◆ 420 °C, line: fitted).

Tables 4 and 5, respectively, show the kinetic and deactivation model parameters of the two catalysts. The activation energy was greatest for HDCCR, followed by HDS, and least for HDM, which is more easily carried out during residue hydrogenation. For the HDM reaction, the activation energy of HDV is significantly smaller than that of HDNi, which indicates that vanadium removal is easier than nickel removal, which is consistent with the other literature [31,32].

**Table 4.** Kinetic and deactivation model parameters of the HDM catalyst.

| Parameters | HDCCR | HDS | HDNi | HDV |
|:---:|:---:|:---:|:---:|:---:|
| $k_{0,i}$ | $9.4324 \times 10^6$ | $2.0883 \times 10^5$ | $3.1355 \times 10^5$ | $3.2462 \times 10^5$ |
| $Ea,i/KJ \cdot mol^{-1}$ | 112.83 | 89.01 | 84.05 | 68.76 |
| $n_i$ | 1.199 | 0.83655 | 1.139463 | 1.0058 |
| $\alpha_i$ | 0.082969 | 1.5838 | 6.6116 | 0.93829 |
| $\beta_i$ | 5.27834 | 2.3581 | 2.7420 | 1.6161 |
| $\gamma_i$ | 4.3251 | 0.76 | 1.69649 | 0.015411 |

**Table 5.** Kinetic and deactivation model parameters of the HDS catalyst.

| Parameters | HDCCR | HDS | HDNi | HDV |
|:---:|:---:|:---:|:---:|:---:|
| $k_{0,i}$ | $4.63227 \times 10^5$ | $3.16620 \times 10^5$ | $1.1392 \times 10^5$ | $6.0785 \times 10^4$ |
| $Ea,i/KJ \cdot mol^{-1}$ | 145.25 | 129.19 | 128.83 | 114.40 |
| $n_i$ | 0.98219 | 0.80902 | 1.2696 | 0.794727 |
| $\alpha_i$ | 0.053560 | 0.088090 | 0.114588 | 0.050319 |
| $\beta_i$ | 1.8923 | 1.2493 | 1.854820 | 0.60018 |
| $\gamma_i$ | 0.41896 | 0.054360 | 0.70240 | 19.986 |

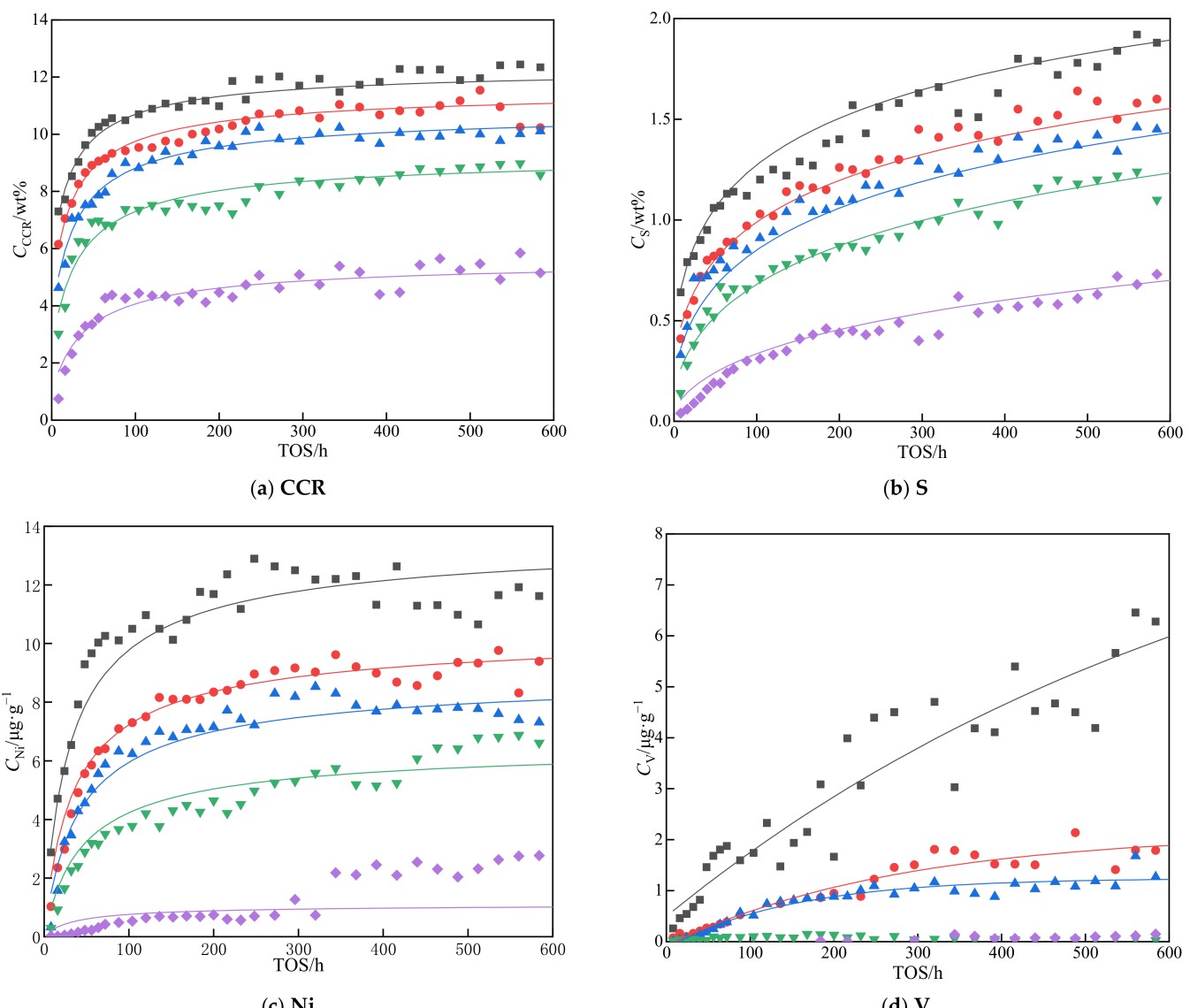

**Figure 8.** Fitting results of the kinetic model for the hydrotreating reaction in the 2nd reactor (■ 405 °C, ● 410 °C, ▲ 415 °C, ▼ 420 °C, and ◆ 430 °C, line: fitted).

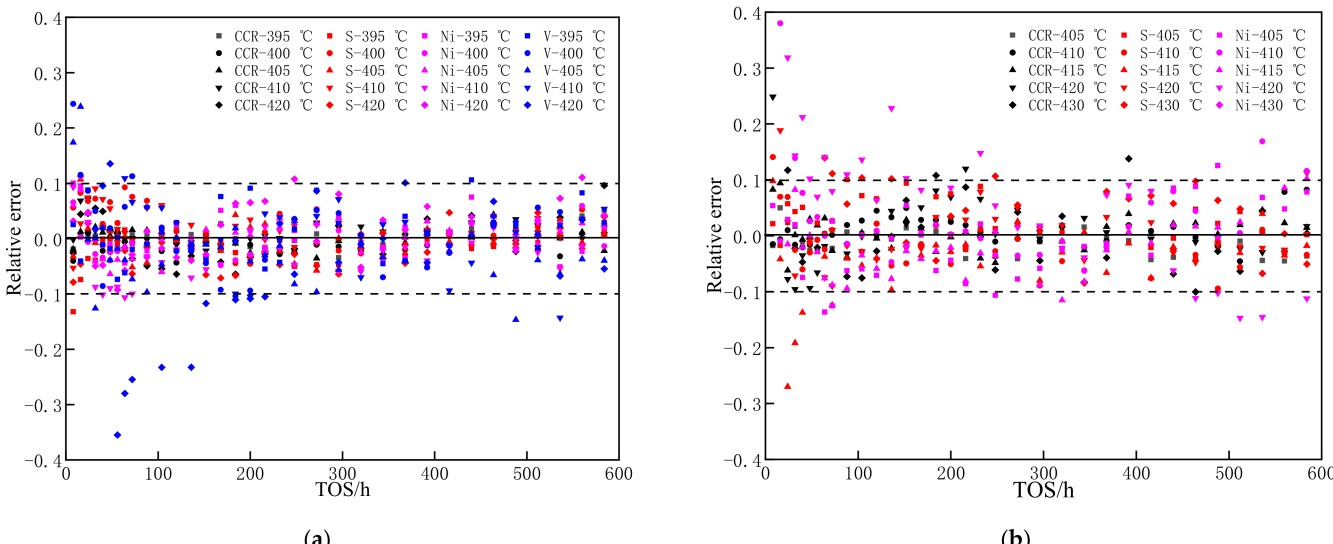

**Figure 9.** Relative error distribution of the kinetic model fitting for the residue hydrotreating reactions. (**a**) The 1st reactor; (**b**) the 2nd reactor (±10%).

The degree of catalyst deactivation can be represented by $\alpha_i$, $\beta_i$, and $\gamma_i$. Here, $\alpha_i$ can indicate the deactivation rate of the catalyst (the larger the $\alpha_i$, the faster the deactivation rate), $\beta_i$ is the order of deactivation, and $\gamma_i$ is the degree of influence of metal deposition on deactivation. Comparing the ai values of the two catalysts in different hydrogenation reactions, it is found that the $\alpha_i$ values of the HDM catalysts are generally higher than those of the HDS catalysts, which indicates that the deactivation rate of the demetallized catalyst is faster. It can also be seen that the $\gamma_i$ values for the HDNi reaction are larger in both reactors, indicating that metal deposition has a greater effect on the HDNi reaction.

The deactivation curves of HDCCR, HDS, HDNi, and HDV in the two reactors can be obtained according to the model parameters, as shown in Figure 10. In the first reactor, the reactivity decreases rapidly within 100 h. The activity of HDCCR, HDS, and HDNi decreases to 60%, and HDV activity decreases to 80%. In the second reactor, the reactions of HDCCR and HDNi decreased rapidly within 200 h, and HDV and HDNi decreased slowly within 600 h. Combined with the characterization of the catalyst, the deposition of coke over 100 h resulted in a reduction in the catalyst active area and pore volume, which was the main reason for the sharp decrease in catalyst activity, after which the deposition of coke reached equilibrium and the slower sustained deactivation was caused by the deposited metal. In the first reactor, the vanadium conversion is high, so at the entrance of the second reactor, the concentration of HDV is low and the reaction occurs mostly in the outer layer of the catalyst, which is less affected by catalyst deactivation, resulting in an almost linear decay of the reactivity. The HDS catalyst could still maintain a high activity in the second reactor, indicating a good activity for sulfur removal.

Comparing the activity of each catalyst in the two reactors shows that the catalyst activity in the second reactor is higher than that of the first reactor, indicating that the deactivation of the HDM catalyst is faster than that of the HDS catalyst, which is due to the removal of impurities in the first reactor. The catalyst deactivation in the fixed-bed reactor is strongly influenced by the composition of the feedstock.

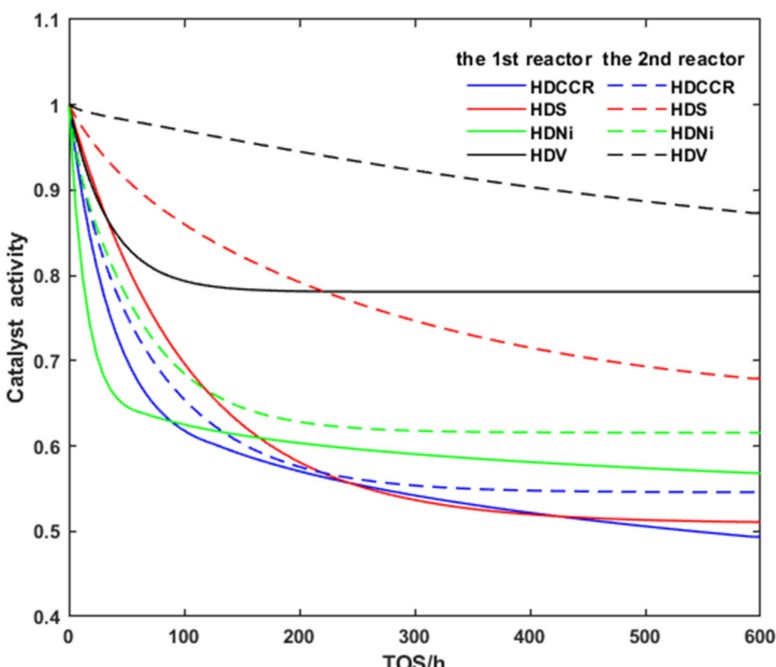

**Figure 10.** Variation of catalyst activity with TOS.

## 4. Conclusions

An empirical model was proposed and the deactivation model parameters of HDCCR, HDS, HDNi, and HDV in two series reactors were obtained by fitting the experimental data at the outlet of the two reactors. The results show that the relative average error of the model does not exceed 5%.

The model parameters indicate that the HDV activation energy is small, and that vanadium can be easily removed. In comparison, the HDCCR activation energy is high, indicating that increasing the temperature favors its proceeding. The HDNi activity is significantly influenced by metal deposition.

The results of this study contribute to a better understanding of residue hydrogenation catalyst deactivation, and the catalyst deactivation model can be used for the optimization of the hydrogenation process and the prediction of hydrogenation results.

**Author Contributions:** Conceptualization and methodology, Z.Z. and H.G.; sample analysis, H.Z. and T.Y.; writing—original draft preparation, K.C. and Y.C.; writing—review and editing, Z.C. All authors have read and agreed to the published version of the manuscript.

**Funding:** This research was funded by the National Key R&D Program of China (No. 2019YFC1906705).

**Institutional Review Board Statement:** Not applicable.

**Informed Consent Statement:** Not applicable.

**Data Availability Statement:** All data collected in this study are contained within the article.

**Conflicts of Interest:** The authors declare no conflict of interest.

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
