# Peer review of "Experimental Determination on the Deactivation Kinetics of Residue Hydroprocessing in a Two-Stage Fixed Bed Loaded with HDM and HDS Catalysts"

_processes, doi:10.3390/pr10091822_

Round 1
Reviewer 1 Report
This study reports the catalyst deactivation due to HDM reaction during hydroprocessing of heavy feed. The experimental data were used to develop a deactivation model based on time-on-stream. The topic deserves publication after addressing the following major comments:
1. Introduction.
The state-of-the-art has not been properly reviewed. Relavant refereces dealing with the same topic were neither commented nor compared with, such as: Fuel 234 (15 December) (2018) 326-334; Fuel 138 (2014) 45-51; Catalysis Today 220– 222 (2014) 221-227; Fuel Processing Technology 123 (2014) 114–121; Fuel 100 (2012) 193-199; Fuel 100 (2012) 17-23; Energy Fuels 21(2007) 636-639; Energy & Fuels 17 (2003) 462-467; Energy & Fuels 16 (2002) 1438-1443.
2. Experimental.
Metal content of catalyst must be provided in table 1.
More properties of the feed must be reported in table 2, such as API gravity, SARA analysis, distillation curve.
The authors commented that "part" (how much?) of the catalyst was unloaded (from where, top or bottom) from the reactor for characterization. Are they aware that doing so the effective amount of catalyst for reaction is diminished, and also if samples are taked from bottom or top of reactor, the coke and metal content will be quite different as funtion of reactor lenght? This is a major drawback that need to be cleary commented in the manuscript.
3. Results and discussion.
No comments are made on the possible mass transfer limitation. The authors must, at least, verified the magnitude of such effects. To do so, please refer to: Applied Catalysis A: General 355 (2009) 1–19
Metals content on spent catalyst must be calculated on a fresh-bases. To do so, please refer to: Fuel 88 (2009) 2311–2314
Textural analysis must be used to determine a semi-quantitative deactivation. To do so, please refer to: Catalysis Communications 133 (5 January) (2020) 105823
The developed deactivation model is based on TOS, however time is not the parameter that affect deactivation but metals and coke deposition. The authors must deeply comment on what they have used TOS instead of coke and metal deposition.
Model parameters need to be clearly assured that they are the optimal ones by sensitivity analysis. To do so, please refer to: Fuel 267 (1 May) (2020) 117113.
All data used for development of deactivation model (variation of sulfur, metals, CCR, and other contents and properties against time) and spent catalyst characterizations must be provided in supplementary information.
Reviewer 2 Report
This is a very interesting piece of work with plenty of results. The article concerns deactivation studies of catalysts used in hydrotreating of residues, which is an area of great interest for the Oil&Gas industry. The article is clear, well-written and conclusions are drawn based on facts and data. The article may be accepted provided minor modifications are carried out.
Authors are kindly requested to provide us with the following enlightenments.
1 – What type of alumina was used in the preparation of the catalysts? It is clear that these are commercial catalysts, however XRD diffraction analysis could give an indication of the alumina type. Deactivation is often related to the type of alumina used as a support.
2 – Why was the microporosity of the catalysts not determined? Deactivation may be caused by micropore fouling. A t-plot analysis could be performed. Please comment on that.
3 – An analysis of figure 10 indicates that deactivation is not homogeneous. There is a steep drop in conversion at the beginning of the operation. Should deactivation occur in a homogeneous manner, such pronounced activity decay would not be observed. Such curve shape is an indication that different mechanisms of deactivation are taking place. Apparently, more active sites are undergoing deactivation first. Please add comments on this topic.
4 – A linear decay is observed for HDV in Figure 10. As previously mentioned, this is a typical homogeneous deactivation curve, in which sites equally undergo deactivation. Since such curve is completely different from the other curves, one must assume that a different deactivation mechanism is taking place. Please comment on that.
Round 2
Reviewer 1 Report
The authors have not taken seriously the comments made in the first round of review. Some recommendations were not properly taken into consideration, such as:
1) Verification of extend of mass transfer limitations by theoretical calculations. A reference was suggested to do this.
2) Calculation and discussion of fresh-based catalyst compositions. Calculations were done but not added in the manuscript with proper discussion and reference provided.
3) Semi-quantitative calculation of catalyst deactivation. Nothing was mentioned nor discussed in the manuscript.
4) Detailed data were suggested to add as supplementary information. The authors refuse to add this information, which would be of help for the readers.
A new round of review is mandatory to verify that these comments are properly addressed.
